# Angiotensin II Promotes Osteocyte RANKL Expression via AT1R Activation

**DOI:** 10.3390/biomedicines13020426

**Published:** 2025-02-10

**Authors:** Jiayi Ren, Aseel Marahleh, Jinghan Ma, Fumitoshi Ohori, Takahiro Noguchi, Ziqiu Fan, Jin Hu, Kohei Narita, Angyi Lin, Hideki Kitaura

**Affiliations:** 1Department of Orthodontics and Dentofacial Orthopedics, Tohoku University Graduate School of Dentistry, Aoba-ku, Sendai 980-8575, Miyagi, Japan; ren.jiayi.p7@dc.tohoku.ac.jp (J.R.); marahleh@tohoku.ac.jp (A.M.); ma.jinghan.c1@tohoku.ac.jp (J.M.); fumitoshi.ohori.b4@tohoku.ac.jp (F.O.); takahiro.noguchi.d4@tohoku.ac.jp (T.N.); fan.ziqiu.q1@dc.tohoku.ac.jp (Z.F.); hu.jin.q5@dc.tohoku.ac.jp (J.H.); kohei.narita.a2@tohoku.ac.jp (K.N.); lin.angyi.r5@dc.tohoku.ac.jp (A.L.); 2Frontier Research Institute for Interdisciplinary Sciences, Tohoku University, Sendai 980-8578, Miyagi, Japan

**Keywords:** angiotensin II, osteocyte, MLO-Y4, osteoclastogenesis

## Abstract

**Background/Objective:** Osteocytes are the most abundant cell type in the skeleton, with key endocrine functions, particularly in regulating osteoblast and osteoclast activity to maintain bone quality. Angiotensin II (Ang II), a critical component of the renin–angiotensin–aldosterone system, is well-known for its role in vasoconstriction during hypertension. Beyond its cardiovascular functions, Ang II participates in various biological processes, including bone metabolism. While its influence on osteoblast proliferation, differentiation, and osteoclastogenesis has been documented, its effects on osteocytes remain unexplored. This study hypothesized that Ang II enhances the osteoclastogenic activity of osteocytes. **Methods:** Mouse calvariae were cultured ex vivo in an Ang II-containing medium, analyzed via immunohistochemistry, and evaluated for osteoclastogenic gene expression through real-time PCR. Western blotting was employed to assess protein levels and signaling pathway activation in the MLO-Y4 osteocytic cell line in vitro. **Results:** Ang II significantly increased the expression of receptor activator of nuclear factor κB ligand (RANKL) and macrophage colony-stimulating factor (M-CSF). These effects were abrogated by azilsartan, a blocker targeting Ang II type 1 receptors (AT1R). p38 and ERK1/2 in the MAPK pathway were also activated by Ang II. **Conclusions:** Ang II enhances osteocyte-mediated osteoclastogenesis via AT1R activation, highlighting its potential as a therapeutic target for bone diseases.

## 1. Introduction

Osteocytes constitute more than 90% of the cells in the skeletal system and are embedded within lacunae in the bone matrix. These cells play crucial roles in transmitting mechanical signals and communicating through their dendritic processes, which help maintain bone quality. Recently, researchers have focused on the functions of osteocytes beyond their mechanotransductive roles, particularly their endocrine functions. Importantly, osteocytes regulate bone metabolism by secreting essential molecules, such as receptor activator of nuclear factor κB ligand (RANKL) and osteoprotegerin (OPG), which control osteoclast activation [1,2,3]. RANKL, essential for osteoclast differentiation, binds to RANK on the surface of osteoclast precursors, promoting the fusion of these precursors into multinucleated osteoclasts that break down and absorb bone tissue. OPG, a decoy receptor for soluble RANKL, inhibits its osteoclastogenic effects [4]. Beyond osteoclast activation, osteocytes express osteocalcin (OCN), fibroblast growth factor 23 (FGF23), dentin matrix protein 1 (DMP1), and sclerostin (SOST), which not only mediate bone turnover but also contribute to systemic energy metabolism and neurodegeneration [2,5,6,7]. Additionally, osteocytes express lipocalin 2, a bone-derived signal that regulates bone homeostasis and appetite, in vitro [8,9]. Consequently, the broader roles of osteocytes in bone biology are increasingly recognized. However, primary osteocyte studies face significant challenges due to difficulties in osteocyte isolation and a high rate of osteocyte death in vitro. Ex vivo culture may be a feasible method for studying primary osteocytes based on the longer life span of osteocytes embedded in bone lacunae [10].

Angiotensin II (Ang II), an octapeptide in the renin–angiotensin–aldosterone system (RAAS), is a pleiotropic peptide that mediates diverse cellular responses beyond its established role as a vasoconstrictor through signaling pathways activated via its type I receptor (AT1R). The classical RAAS involves renin produced by the kidney, which converts angiotensinogen from the liver into Ang I. This is followed by Ang I cleavage by ACE on lung cell surfaces to form circulating Ang II. Over the past few decades, studies have shown that Ang II not only constricts blood vessels but also contributes to inflammation, fibrosis, and cancer metastasis [11,12,13,14]. Furthermore, extensive research has uncovered the complexity of RAAS endocrine regulation [15]. This includes the discovery of additional enzymes, short-chain peptide products of angiotensin, and localized RAAS components in tissues, which have broadened the understanding of RAAS functions beyond its traditional framework [16].

In the context of bone metabolism, the relationship between bone metabolism and hypertension in humans remains incompletely understood [17,18,19]. Metabolic syndrome, which encompasses hypertension, is thought to negatively affect bone health [20,21]. Antihypertensive drugs targeting the Ang II/AT1R pathway, such as angiotensin-converting enzyme (ACE) inhibitors and angiotensin receptor blockers (ARBs), have been found to improve bone quality and reduce bone resorption [22,23]. These findings suggest that Ang II promotes bone resorption. Evidence further indicates that Ang II contributes to osteoporosis by enhancing osteoclast activation and degrading the bone matrix [24]. Ang II can also stimulate RANKL production in osteoblasts, thereby augmenting osteoclast differentiation and activity [24,25]. However, despite considerable evidence that Ang II influences bone turnover toward resorption, its effects on osteocytes, the most abundant bone cells, remain unclear [26].

In this study, we investigated the effects of Ang II on osteocyte activity and its capacity to induce osteoclastogenesis. Through ex vivo and in vitro studies, we aimed to bridge the gap between RAAS and osteocyte function, providing a foundation for future research into therapeutic targets for bone diseases under physiological and pathological conditions involving RAAS.

## 2. Materials and Methods

### 2.1. Animals and Reagents

Eight-week-old male C57BL/6J mice were purchased from CLEA (Tokyo, Japan). All animal care and experimental procedures complied with the Regulations for Animal Experiments of Tohoku University (certification of conformity: no. 2018DnA-049-12). Angiotensin II and azilsartan (Sigma-Aldrich, St. Louis, MO, USA) were used in this study.

### 2.2. Bone Preparation

The mice were euthanized by cervical dislocation immediately after being anesthetized with isoflurane inhalation (Pfizer, New York, NY, USA) under the fume hood. The hair and skin were sprayed with 70% ethanol, and the calvarial tissue, excluding the sagittal suture, was removed using a 4 mm biopsy punch (KAI Medical, Tokyo, Japan) to minimize contamination with mesenchymal cells (Figure 1A), as described by Bellido and Delgado-Calle [27]. Each calvarial piece was bisected into equally sized semicircles (Figure 1B). The bones were then fractionated five times using collagenase and ethylenediaminetetraacetic acid (EDTA), as described previously [28]. Briefly, 2 mg/mL collagenase dissolved in an isolation buffer sterilized through a 0.2 µm strainer was used for fractions 1–3 and 5, while 5 mM EDTA in PBS containing 0.1% BSA was used for fraction 4. Fractionation was performed in a BioShaker (TAITEC, Saitama, Japan) at 210 rpm for 25 min at 37 °C. The supernatant from all fractions was discarded, and the bone pieces were randomly divided into groups according to the experimental design.

### 2.3. Immunohistochemistry

The calvariae semicircles were randomly divided into the control group (0 days) and experimental groups (1 day and 3 days), with 3 pieces per group, and cultured in 24-well plates at 37 °C in a 5% CO_2_ humidified incubator. Minimum essential medium α (α-MEM) supplemented with 10% fetal bovine serum (FBS), 100 U/mL penicillin, and 100 μg/mL streptomycin (Wako, Osaka, Japan) was used, with 10^−^⁷ M Ang II or PBS. At 0 (without treatment), 1, and 3 days, the calvariae were collected, fixed in 10% formalin at 4 °C, and decalcified with 14% EDTA at room temperature for 5 days. The decalcified calvariae were dehydrated, embedded in paraffin, and sectioned into 5 µm slices (Figure 1B). After deparaffinization, antigens were retrieved in 1 mM EDTA (pH 8) at 60 °C overnight. The sections were incubated in 3% H₂O₂ in water to block endogenous peroxidase activity. Following a rinse, the sections were blocked with 5% skimmed milk containing 0.5% Triton at room temperature for 1 h. The samples were then incubated with an anti-RANKL antibody (ab216484, 1:400, Abcam, Cambridge, UK) overnight at 4 °C and rinsed with a buffer. The control slides for each sample were incubated with a blocking buffer instead of the primary antibody. The VECTASTAIN Elite ABC Kit (VEC, Newark, CA, USA) was used for secondary antibody incubation and signal enhancement. The DAB substrate was applied for detection, and the nuclei were counterstained with hematoxylin. The sections were dehydrated and cleaned for observation and analysis [29,30] (Figure 1A).

### 2.4. Cell Viability and Cytotoxicity Assay

MLO-Y4 cells were purchased from AddexBio Technologies (San Diego, CA, USA). A total of 1 × 10⁴ cells per well were seeded in a 96-well plate and treated with a concentration gradient of Ang II (10^−^⁸ M to 10^−^⁶ M) for 24 or 48 h. CCK-8 solution (10 µL; DOJINDO, Kumamoto, Japan) was added to 100 µL of the cell culture medium and incubated at 37 °C for 1–2 h in the dark. Then, 100 µL of the supernatant medium was mixed with 100 µL of a pre-mixed LDH assay working solution from the same kit. After incubation for 30 min at room temperature, the reactions were stopped using the stop solution (50 µL). The absorbance at 450 nm was measured using a microplate reader (Sunrise™ Remote, Tecan, Männedorf, Switzerland) with the LS-Plate Manager 2004 software (version 3.00). Wells containing only the medium were used as blank controls.

### 2.5. RNA Extraction and Real-Time Reverse Transcription–Polymerase Chain Reaction

The cells (5 × 10⁴ per well) seeded in a 24-well plate were treated with 10^−^⁷ M Ang II and 10^−^⁶ M azilsartan and cultured for 48 h. The RNA extraction followed the manufacturer’s protocol using RNeasy Kits and Qiagen shredders (QIAGEN, Hilden, Germany). The RNA concentration and quality were assessed using a nanophotometer (Implen, Munich, Germany). Complementary DNA (cDNA) was synthesized using the Superscript IV Reverse Transcriptase System (Invitrogen, Thermo Fisher Scientific, Waltham, MA, USA) with oligo dT primers. The PCR program included an initial step at 95 °C for 30 s, followed by 39 cycles of denaturation at 95 °C for 5 s and annealing/elongation at 60 °C for 30 s. A Bio-Rad CFX96 Touch Real-Time PCR Detection System and TB Green^®^ Premix Ex Taq™ II (TakaraBio Inc., Shiga, Japan) were used for the reactions. The following primers were used: 5′-GGTGGAGCCAAAAGGGTCA (forward) and 5′-GGGGGCTAAGCAGTTGGT (reverse) for GAPDH; 5′-CCTGAGGCCCAGCCATTT (forward) and 5′-CTTGGCCCAGCCTCGAT (reverse) for RANKL; 5′-ATCAGAGCCTCATCACCTT (forward) and 5′-CTTAGGTCCAACTACAGAGGAAC (reverse) for OPG; 5′-TGATTGGGAATGGACACCTG (forward) and 5′-AAAGGCAATCTGGCATGAAGT (reverse) for M-CSF; and 5′-AGTCGCACTCAAGCCTGTCT (forward) and 5′-ACTGGTCCTTTGGTCGTGAG (reverse) for AT1R.

### 2.6. Western Blotting

MLO-Y4 cells were seeded in a 12-well plate at a density of 2.5 × 10^5^/well in 10% FBS medium and cultured overnight at 37 °C in 5% CO_2_. Serum restriction was performed by culture in 2% FBS medium for 12 h and then changed to 1% FBS medium. For the signaling cascade study, cells were stimulated by 10^−7^ M Ang II in 1% FBS for 0, 5 min, 15 min, and 30 min. In the protein production study, cells were stimulated by 10^−7^ M Ang II in 1% FBS for 24 h. Protein samples were isolated by radioimmunoprecipitation assay buffer. The lysates were centrifuged in QIAGEN shredders for homogenization and cell pellet removal. The protein samples were reduced by 2.5% *v*/*v* 2-mercaptoethanol and mixed with Laemmli sample buffer (Bio-Rad Laboratories Inc., Hercules, CA, USA). After being denatured by boiling at 95 °C for 5 min, the protein samples were loaded on Mini-PROTEAN^®^ TGX™ Precast Gels (Bio-Rad) for SDS-PAGE. The protein on the gel was transferred to a PVDF membrane (Bio-Rad). Before primary antibody incubation, the membranes were blocked by Block Ace (Yukijirushi, Sapporo, Japan) in TBS-T with sodium azide, which was used for preservation. Anti-RANKL (ab216484, 1:3000, Abcam), Agtr1polyclonal antibody (25343-1-AP, 1:1000; Proteintech Group, Inc., Chicago, IL, USA), phospho-SAPK/JNK (Thr183/Tyr185) (98F2) rabbit mAb (1:3000), phospho-p44/42 MAPK (Erk1/2) (Thr202/Tyr204) antibody (1:3000), phospho-p38 MAPK (Thr180/Tyr182) (D3F9) XP rabbit mAb (1:3000), phospho-IκBα (Ser32) (14D4) rabbit mAb #2859 (1:3000), IκBα antibody #9242 (1:5000, Cell Signaling Technology, Danvers, MA, USA), and beta-actin monoclonal antibody (1:5000) were incubated with membranes overnight at 4 °C. After being rinsed with TBS-T to remove extra primary antibodies, the membranes were incubated with HRP-conjugated anti-rabbit secondary antibody (1:5000–10,000, Cell Signaling Technology) or anti-mouse antibody (1:10,000) for 1 h at room temperature, and chemical blot was detected with SuperSignal™ West Femto Maximum Sensitivity Substrate (Thermo Fisher Scientific) [31].

### 2.7. Statistical Analysis

For the IHC analysis, five successfully stained stripes of one group were randomly selected for whole-stripe-segmented imaging at the same magnitude with minimum overlap. All images were evaluated twice manually and independently, and the average positive cell percentage per image, after subtracting the control slide average, in each group was analyzed using one-way ANOVA. All data are expressed as means ± standard deviation. To assess group differences, unpaired and paired t-tests were used in the WB evaluation, and one-way ANOVA with Tukey’s post hoc test was used for IHC-positive cell percentages, cell viability measurements, and real-time PCR analysis, with statistical significance defined as *p* < 0.05. Data normality was assessed by the Shapiro–Wilk test, and data were considered normally distributed if the *p*-values were greater than 0.05. The paired *t*-test was conducted when the hypothesis of insignificant variance differences was supported. The data analysis and visualization were performed using GraphPad Prism (version 10.3.1).

## 3. Results

### 3.1. Multistep Fractionation Ensured Purity of Cell Types in Ex Vivo Culture

Multiple fractionation cycles were performed to eliminate osteoblasts and fibroblasts from the bone surface, and suspended cells were observed in each fraction. A gradual decrease in the number of suspended cells was noted during the progressive dissociation of non-osteocytes. By the fifth fraction, suspended cells were scarcely visible, while the bone fragments remained intact with clearly distinguishable lacunae (Figure 1C). Tissue section analysis confirmed the complete removal of superficial cells, leaving predominantly osteocytes in the bone matrix (Figure 2A).

### 3.2. Ang II Impacted RANKL Expression in Ex Vivo-Cultured Osteocytes

We performed tissue sectioning and IHC staining of the calvarial pieces after ex vivo culture. The percentage of RANKL detection in osteocytes was elevated in the Ang II-treated group compared with the control group after 1- and 3-day cultures (Figure 2B,C). After 1 day, RANKL detection was significantly increased in the Ang II group. After 3 days of culture, although the percentage of RANKL-positive osteocytes increased in the Ang II group, the proportion of nuclear-stained cells relative to the total lacunae decreased. This suggests that some osteocytes died after three days of ex vivo culture.

### 3.3. MLO-Y4 Cell Viability and Ang II Cytotoxicity

We tested Ang II concentrations ranging from 10^−^⁸ M to 10^−^⁶ M and used the CCK-8 assay to assess MLO-Y4 cell viability, as well as the LDH assay kit-WST to evaluate cytotoxicity. Ang II at these concentrations did not affect cell viability for up to 48 h of stimulation (Figure 3A). Similarly, no cytotoxicity was observed at 24 h. However, at 48 h, 10^−^⁶ M Ang II exhibited slight cytotoxicity compared with 10^−^⁷ M, though no significant difference was noted between the negative control and 10^−^⁶ M (Figure 3B). To minimize additional confounding factors, we selected 10^−^⁷ M as the optimal concentration for subsequent experiments.

### 3.4. Ang II Increased Osteoclastogenic Gene Expression via AT1R

In MLO-Y4 cells cultured with Ang II, the mRNA expression of RANKL, macrophage colony-stimulating factor (M-CSF), and AT1R increased by approximately 1.5-fold. The AT1R blocker azilsartan significantly inhibited the upregulation of these genes (Figure 4A–C). OPG expression was unaffected by Ang II or azilsartan, resulting in an elevated RANKL/OPG ratio induced by Ang II, which was reduced by azilsartan (Figure 4D,E). These results indicate that Ang II activates the AT1 receptor on osteocytes to promote RANKL and M-CSF expression.

### 3.5. Ang II Increased RANKL Protein Level and Activated MAPK Pathway

In serum-restricted MLO-Y4 cells stimulated with Ang II, the protein level of RANKL was significantly increased. The production of AT1R was also increased, though not significantly, which was consistent with the mRNA results (Appendix A). Regarding the signaling cascade, the MAPK and NF-κB pathways were investigated. The phosphorylation of p38 and ERK1/2 in the MAPK pathway was promoted by Ang II, whereas JNK/MAPK and the NF-κB pathway were not activated by Ang II (Figure 5 and Appendix A). These results indicate that the promotion of RANKL expression by Ang II is possibly regulated via the activation of p38 and ERK in the MAPK pathway.

## 4. Discussion

Ang II has been recognized for its osteoclastogenic properties for nearly three decades. Hatton et al. identified the local presence of RAAS components in bone cells and demonstrated that Ang II stimulates bone resorption in co-cultures of osteoclasts and bone slices, expected to have osteoblasts present [24]. Shimizu et al. showed that Ang II promotes osteoporosis in wild-type ovariectomized (OVX) mice, while the same procedure in AT1R-knockout (KO) mice did not result in reduced bone quality. Furthermore, their study revealed that Ang II induced osteoclast formation in co-cultures of osteoblasts and osteoclast precursors, accompanied by increased RANKL expression in osteoblasts [25]. However, both studies independently noted that Ang II does not stimulate the differentiation of pure osteoclast precursors into osteoclasts. Tsukuba hypertensive mice, which express both human renin and angiotensinogen genes, resulting in high Ang II production, exhibit low bone mass, which can be improved through ACE inhibition [32]. Recent research has also investigated the effects of Ang II and its blockers on bone lesions in animal models of arthritis [33,34].

However, a gap remains between Ang II and osteocytes, both of which are important in regulating osteoclastogenesis. Osteocytes modulate bone turnover by expressing positive factors, such as OPG, OCN, and FGF23, and negative factors, such as RANKL and SOST. The RANK/RANKL/OPG interaction is the most crucial and direct pathway for osteoclastogenesis [35]. M-CSF is another key molecule that induces bone marrow cells to differentiate into osteoclast precursors, also known as bone marrow-derived macrophages (BMMs). It also helps maintain the characteristics of BMMs and accelerates their proliferation during osteoclastogenesis [36]. To investigate whether Ang II affects the osteoclast activation capacity of osteocytes, we cultured mouse calvariae ex vivo with Ang II for 1 and 3 days. Using IHC, we found that the RANKL-positive osteocyte ratio increased after 1 day compared with 0 and 3 days. A decrease in cell survival was also observed after 3 days of culture. But this is not the only case. Pathak and colleagues reported that the survival rate of osteocytes after a 7-day ex vivo culture was around 60% [10]. These results suggest a promotive role of Ang II in osteocyte osteoclastogenic potential.

MLO-Y4 is an early osteocyte cell line derived from mouse long-bone osteocytes [37]. Although it may not be suitable for osteoblast function research due to its low expression of SOST and FGF23, it is ideal for osteoclastogenesis-related studies [38]. MLO-Y4 has been reported to independently support osteoclast formation at very low densities without any additional stimuli while also expressing and secreting large amounts of RANKL and M-CSF [39]. Controversially, mechanical loading is an inhibitor of osteocyte osteoclastogenesis due to the upregulated OPG level [40]. To explore the effect of Ang II on the osteoclastogenic effort of osteocytes, we started by assessing MLO-Y4 cell viability and toxicity following Ang II treatment at concentrations ranging from 10^−8^ to 10^−6^ M. The CCK-8 and LDH assay results showed no significant changes after 24 h of stimulation. After 48 h, although cell viability remained unchanged, 10^−6^ M Ang II treatment caused significant cytotoxicity compared with 10^−7^ M. The discrepancy between the CCK-8 and LDH results may be attributed to their different measurement principles. The CCK-8 assay quantifies total dehydrogenase activity in viable cells, whereas the LDH assay detects the release of LDH into the culture medium when the plasma membrane is damaged. The 10^−6^ M concentration may cause increased membrane permeability, leading to excessive LDH leakage without inducing cell death. This condition did not significantly impact cell viability, as measured by the CCK-8 assay.

For functional validation results, at the transcript level, Ang II (10^−7^ M) increased the mRNA expression of osteoclastogenic factors RANKL and M-CSF, along with an increase in AT1R expression in MLO-Y4 cells. The OPG level was not affected by Ang II. These interesting results indicate that Ang II (10^−7^ M) not only directly enhances osteocyte support for osteoclast differentiation but also indirectly maintains the proliferation and survival of osteoclast precursors (BMMs). At the translational level, RANKL expression was also upregulated by Ang II. Since the MAPK and NF-κB pathways lead to RANKL expression, we demonstrated that Ang II activated the phosphorylation of p38 and ERK1/2 in the MAPK pathway, which was also confirmed by Zhang and colleagues using synoviocytes [41]. Although their study showed a significant elevation in AT1R protein levels induced by Ang II, the higher concentration of Ang II they used (10^−6^ M) might explain the difference. These results provide further insights into how Ang II is closely associated with osteoclastogenesis and bone lesions.

One limitation of this study was its exclusive focus on the regulatory effects on osteoclastogenesis. While the findings offer novel insights into the interaction between the RAAS, osteocytes, and osteoclasts, the effects on osteoblasts or other key factors involved in bone remodeling and systemic metabolism were not extensively explored. Further studies are necessary to investigate the impact of the RAAS on osteocyte-regulated osteoblast activity and the broader hormonal and molecular mechanisms involved. Such research could provide a more comprehensive understanding of how the RAAS mediates bone turnover balance through osteocytes and help clarify the connection between hypertension and osteoporosis.

In summary, our study addresses a critical gap in understanding the relationship between the RAAS and osteocytes by examining the impact of Ang II and its receptor, AT1R, on osteocyte activity and osteoclastogenic function in both ex vivo and in vitro models. By demonstrating how Ang II affects osteocyte-derived osteoclastogenic factors, this study establishes a novel link between the RAAS and bone cell interactions. These findings contribute to the growing body of knowledge on bone metabolism and may pave the way for the development of therapeutic strategies targeting RAAS-related pathways in osteolytic diseases.

## 5. Conclusions

The present study demonstrated that Ang II induces osteocytes to exhibit osteoclastogenic properties by increasing the expression of RANKL and M-CSF through the AT1 receptor.

## Figures and Tables

**Figure 1 biomedicines-13-00426-f001:**
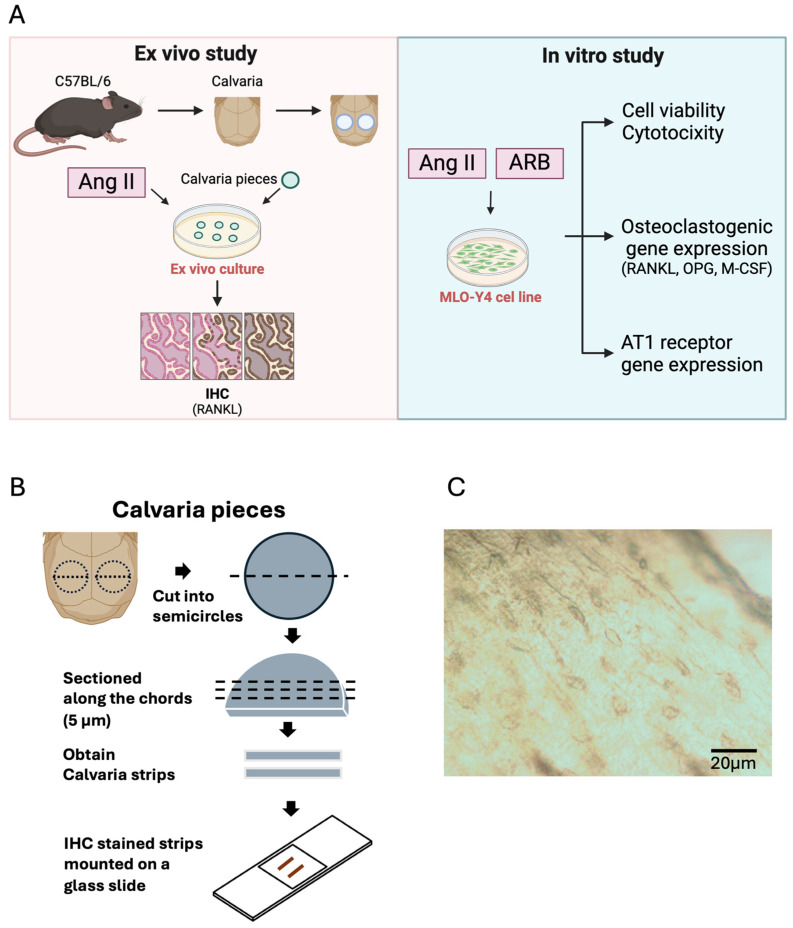
Methodology of this study. (**A**) Schematic representation of the ex vivo and in vitro studies. For the ex vivo study, calvariae excised from wild-type mice were excluded from the mesenchymal suture using a biopsy punch and then cut into semicircles. The calvariae pieces were cultured in a medium containing angiotensin II (Ang II) for 1 and 3 days, followed by fixation, decalcification, embedding, and sectioning. The sections were analyzed via immunohistochemistry (IHC) to examine receptor activator of nuclear factor κB ligand (RANKL) expression. ARB: Ang II type 1 receptor blocker, in this study we used azilsartan. The schematic was created with BioRender. (**B**) Calvariae were cut into semicircles and sectioned along the chords to observe osteocytes within the lacunae. (**C**) Lacunae observed using a light microscope during ex vivo culture. Scale bar = 20 μm.

**Figure 2 biomedicines-13-00426-f002:**
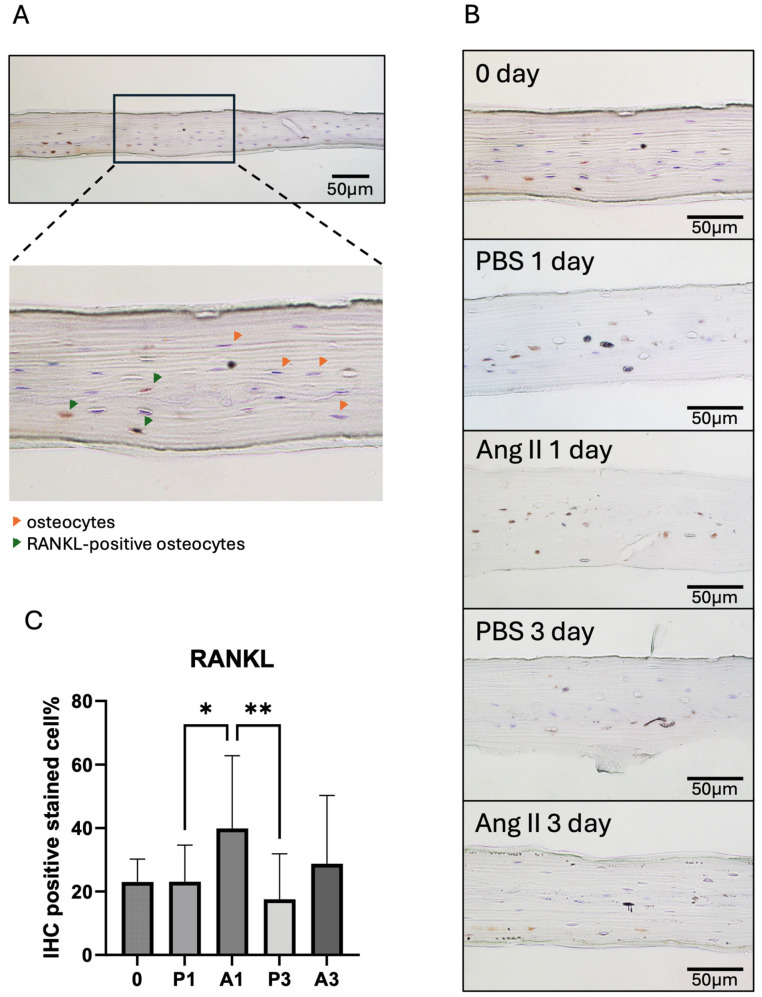
Ang II-containing ex vivo culture increased RANKL expression in osteocytes. (**A**) Illustration of the measurement method. RANKL-positive osteocytes appear yellow to brown. (**B**) Representative images of each culture condition. Scale bar = 50 μm. (**C**) Ratio of RANKL-positive osteocytes relative to hematoxylin-stained nuclei. Data represent the average of over ten valid images per sample. Tukey’s test was used to determine statistical significance between groups (n = 3; * *p* < 0.05 and ** *p* < 0.01).

**Figure 3 biomedicines-13-00426-f003:**
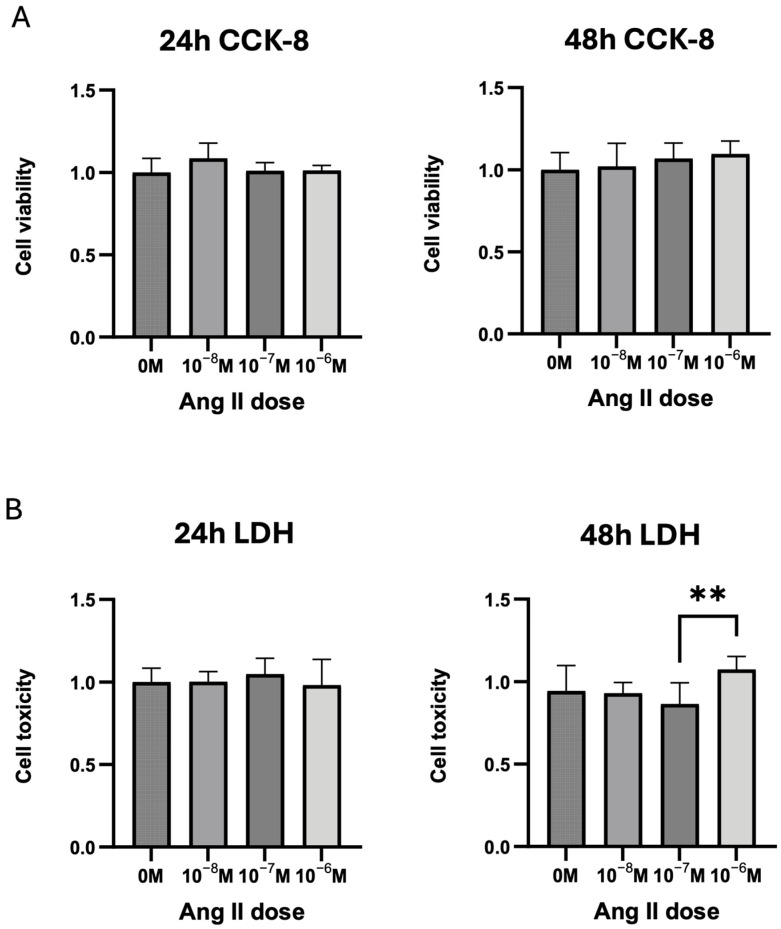
Ang II (10^−^⁸–10^−^⁶ M) did not affect MLO-Y4 cell viability (24 h and 48 h of stimulation) but showed potential cytotoxicity after 48 h of stimulation at 10^−^⁶ M. (**A**) Cell viability assessed using the CCK-8 assay. (**B**) Cytotoxicity evaluated using the lactate dehydrogenase (LDH) assay kit with water-soluble tetrazolium dye (WST) on culture supernatants. Non-significant data are not labeled. Tukey’s test was used to determine statistical significance between groups (n = 4; ** *p* < 0.01).

**Figure 4 biomedicines-13-00426-f004:**
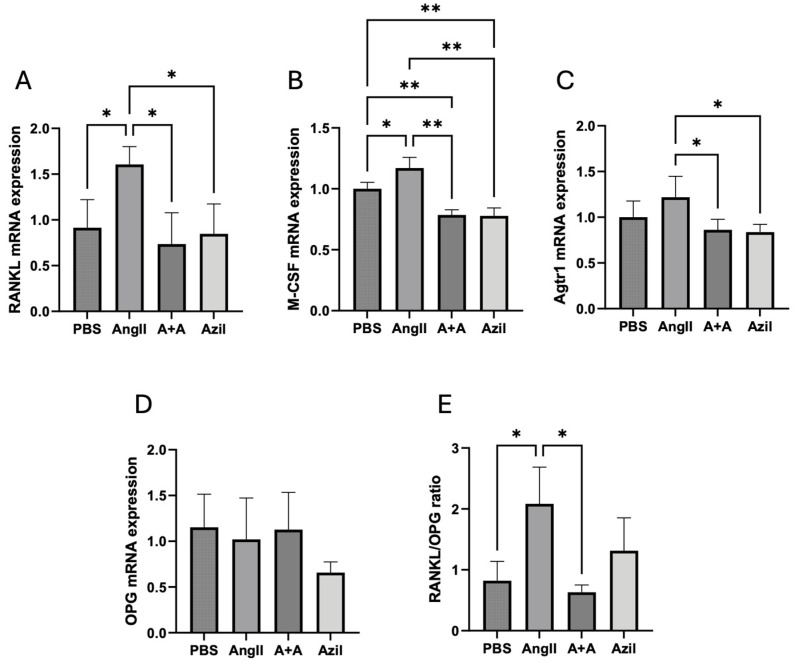
Ang II (10^−^⁷ M) increased the expression of osteoclastogenic genes and Ang II type 1 receptor (AT1R) mRNA levels, while AT1R blockade by azilsartan abolished these effects. (**A**) RANKL mRNA levels. (**B**) Macrophage colony-stimulating factor (M-CSF) mRNA levels. (**C**) AT1R mRNA levels. (**D**) Osteoprotegerin (OPG) mRNA levels. (**E**) RANKL/OPG mRNA ratio. A + A refers to the double administration of Ang II and azilsartan. Non-significant data are not labeled. Tukey’s test was used to determine statistical significance between groups (n = 4; * *p* < 0.05 and ** *p* < 0.01).

**Figure 5 biomedicines-13-00426-f005:**
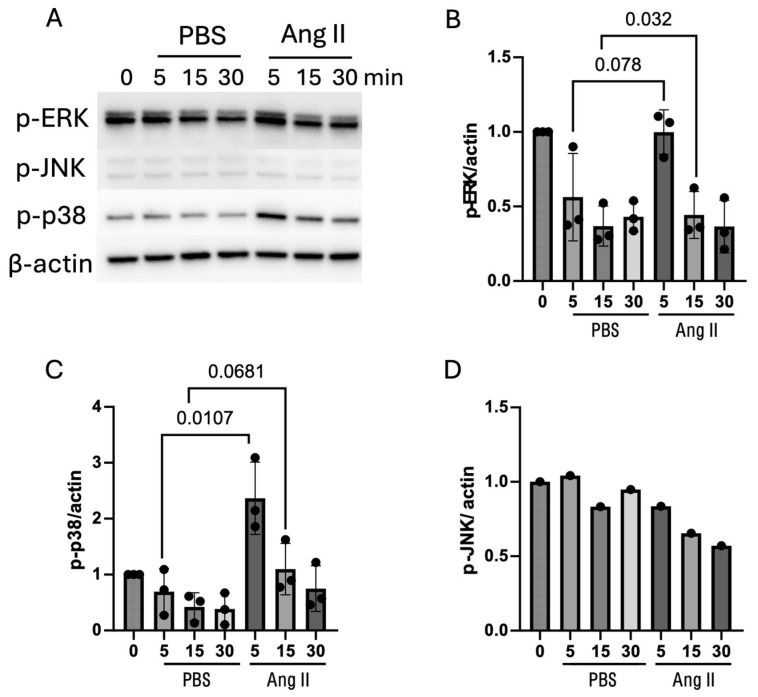
Ang II (10^−^⁷ M) activated the MAPK pathway. (**A**) Western blot images of phosphorylated ERK1/2, JNK, p38, and β-actin. (**B**) Statistical chart of phosphorylated ERK1/2 relative to β-actin. (**C**) Statistical chart of phosphorylated p38 relative to β-actin. (**D**) Statistical chart of phosphorylated JNK relative to β-actin. Paired t-test was used to determine statistical significance between groups (n = 3).

## Data Availability

The original contributions presented in this study are included in the article/Appendix A. Further inquiries can be directed to the corresponding author.

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
