# Peer review of "Angiotensin II Promotes Osteocyte RANKL Expression via AT1R Activation"

_biomedicines, 2025, doi:10.3390/biomedicines13020426_

Round 1
Reviewer 1 Report
Comments and Suggestions for Authors
Dear Authors
Overall, it is a well-written manuscript. The following feedback is provided:
Methods
Line 143-147
2.6 Statistical analysis
The authors have not described how they had confirmed the data was normally distributed; clarify.
Specify what variables were analyzed by ANOVA. For example, calvariae were harvested at 0, 1 and 3 days. How were repeated measures accounted for?
Results
Line 150 – 153. Omit. These statements are an unnecessary repetition of the methods.
Line 176-177. Provide a reference for this statement.
Figures. Legend for each file should fully explain the abbreviations used so that they can be interpreted without reference to the body of the manuscript.
Discussion
Lines 209-221. Omit. These statements are not a discussion of results. They belong in the introduction.
Liness 223-236. So – how do these other studies compare with your results? The discussion section must focus on comparing your results with those of other publications.
Author Response
Response to Reviewer 1 Comments
Thank you very much for taking the time to review this manuscript. Please find the detailed responses below and the corresponding revisions highlighted in the re-submitted files.
Comments 1: Methods
Line 143-147
2.6 Statistical analysis
The authors have not described how they had confirmed the data was normally distributed; clarify.
Specify what variables were analyzed by ANOVA. For example, calvariae were harvested at 0, 1 and 3 days. How were repeated measures accounted for?
Response 1: Thank you for pointing this out. We agree. Therefore, we modified the Methods 2.3 Immunohistochemistry and 2.7 Statistical analysis to clarify the group separating, variables been analyzed and data normality confirmation. (Line 113-126, 187-198)
In our experiment, each mouse provided 2 samples which have been randomly divided into 5 groups (0, P1, P3, A1, A3), we can’t follow which two pieces are from one mouse. Therefore, relative to repeated measure, one-way ANOVA is more appropriate to compare differences among groups that treated as the independent observations.
Comments 2: Results
Line 150 – 153. Omit. These statements are an unnecessary repetition of the methods.
Response 2: Thank you for this comment. We have modified Result 3.1. (Line 202)
Comments 3: Results
Line 176-177. Provide a reference for this statement.
Response 3: Thank you for your comment. This statement is concluded from our previous experience during treating with primary osteocytes, in which condition primary osteocytes usually survive no longer than 1 week after isolation, and this observation is evidenced by Pathak JL and colleagues [1]. In our experiment, we observed a decline of stained nucleus in bone lacunae without statistical measurement, but survived cells should be more than 60%. We revised this statement in Result 3.2 (Line 224) and cited this article in both Introduction and Discussion sections (Line 52 & 299).
[1] Pathak JL, Bakker AD, Luyten FP, Verschueren P, Lems WF, Klein-Nulend J, et al. Systemic inflammation affects human osteocyte-specific protein and cytokine expression. Calcif Tissue Int (2016) 98(6):596–608. doi: 10.1007/s00223-016-0116-8
Comments 4: Figures. Legend for each file should fully explain the abbreviations used so that they can be interpreted without reference to the body of the manuscript.
Response 4: Thank you for pointing this out. We have, accordingly, added the full names of all the abbreviations when there are first mentioned in Figure legends for easier reading. (Line 108, 237-238, 249-252)
Comments 5: Discussion
Lines 209-221. Omit. These statements are not a discussion of results. They belong in the introduction.
Response 5: Thank you for this comment. Agree. Accordingly, we have modified both the Discussion and Introduction around this content. (Line 49-65)
Comments 6: Discussion
Lines 223-236. So – how do these other studies compare with your results? The discussion section must focus on comparing your results with those of other publications.
Response 6: Thank you for your comment. In this paragraph, we aimed to explain how Ang II affects osteoclastogenesis based on previous studies. We agree that a more direct comparison between our results and those of other studies is necessary. Therefore, we have revised the Discussion section, highlighted similarities, differences and potential explanations for any discrepancies. (Line 296-299, 306-308, 321-329)
Response to Reviewer 1 Comments
|

Reviewer 2 Report
Comments and Suggestions for Authors
The authors investigated the effects of Ang II on osteocyte activity and its capacity to induce osteoclastogenesis. This work has certain scientific value and significance. Based on above issue, I suggest making revisions before publishing.
1. The author should supplement WB protein experiments, not just PCR analysis.
2. The main sources and structural characteristics of Ang II should be given a basic introduction.
3. Animal anesthetics and grouping should be elaborated in detail.
4. All "p" should be italicized.
5. Please carefully check all writing errors.
Author Response
Response to Reviewer 2 Comments
Thank you very much for taking the time to review this manuscript. Please find the detailed responses below and the corresponding revisions highlighted in the re-submitted files.
Comments 1: The author should supplement WB protein experiments, not just PCR analysis.
Response 1: Thank you for pointing this out. We agree with this comment. Therefore, we have investigated the protein level of RANKL and Agtr1 by WB and added the results as supplement materials. We also modified the corresponding contents in Methods 2.6, Results 3.4 and Discussion section (Line 161-185, 255-269, 324-329).
Comments 2: The main sources and structural characteristics of Ang II should be given a basic introduction.
Response 2: Thank you for your comment. We have revised it in the Introduction. (Line 53-58)
Comments 3: Animal anesthetics and grouping should be elaborated in detail.
Response 3: Thank you for pointing this out. We have revised the Methods 2.2 to emphasize this point. (Line 89-90, 100-101)
Comments 4: All "p" should be italicized.
Response 4: Thank you for this comment. We have modified the font of “p” to italic.
Comments 5: Please carefully check all writing errors.
Response 5: Thank you for addressing this point. We have carefully checked the writing again.
Reviewer 3 Report
Comments and Suggestions for Authors
This is a straightforward in vitro study to test the effects of angiotensin II on RANKL expression in osteocytes using explant and MLO-Y4 cell line. The authors demonstrated that angiotensin II suppressed RANKL expression in explant, MCSF and RANKL mRNA expression and RANKL/OPG ratio in MLO-Y4 cell line. I feel that to complete the story, the authors need to demonstrate increased osteoclastogenesis and bone resorption in the explant model, but it is understood that the decellularization method has removed the pre-osteoclasts from the bone surface. This limitation could be addressed in the discussion. Other limitations include:
- Fractioned or decellularized bone samples?
- There are no details on unbiased counting of RANKL-positive osteocytes in IHC samples.
- The gene expression results are not validated with protein expression results.
- The exact signalling cascade leading to RANKL expression is not illustrated in this experiment.
Author Response
Response to Reviewer 3 Comments
Thank you very much for taking the time to review this manuscript. Please find the detailed responses below and the corresponding revisions highlighted in the re-submitted files.
Comments 1: Fractioned or decellularized bone samples?
Response 1: Thank you for your comment. The samples should be described as fractioned since we did not decellularized bone samples. Osteocytes and a minimum amount of lining cells remains in the bone samples.
Comments 2: There are no details on unbiased counting of RANKL-positive osteocytes in IHC samples.
Response 2: Thank you for pointing this out. We agree with your advice. Accordingly, we have revised the Methods 2.7 Statistical analysis and provided more details of RANKL-IHC evaluation. (Line 125-126, 187-191)
Comments 3: The gene expression results are not validated with protein expression results.
Comments 4: The exact signalling cascade leading to RANKL expression is not illustrated in this experiment.
Response 3: Thank you for your comments. Since both comments 3 and 4 are relate to protein-level experiments, we would like to address them together. We agree with your suggestions. Accordingly, we have investigated the protein expression level of RANKL and Agtr1, as well as the signaling pathway leading to RANKL expression using Western blot. The results have been incorporated into Results 3.4 and added as supplementary materials. We have also revised the corresponding sections of the manuscript accordingly. (Line 160-185, 255-269, 324-329)

Round 2
Reviewer 2 Report
Comments and Suggestions for Authors
Accept